# *"A friend during troubled times"*: Experiences of family caregivers to persons with dementia when receiving professional support via a mobile app

Åsa Dorell [1] *, Hanne Konradsen [1,2,3], Ana Paula Kallström[1], Zarina Nahar Kabir[1]

**1** Department of Neurobiology, Care Sciences and Society, NVS, Karolinska Institutet, Stockholm, Sweden, **2** Department of Clinical Medicine, Faculty of Health and Medical Sciences, University of Copenhagen, Copenhagen, Denmark, **3** Department of Gastroenterology, Herley and Gentofte Hospital, Herlev, Denmark

* asa.dorell@ki.se

**Data Availability Statement:** All relevant data are within the manuscript and its Supporting Information files.

## Introduction

The number of adults living with dementia is increasing. In 2018, fifty million people worldwide were diagnosed with dementia and it is estimated that this figure will triple by 2050 [1]. The number of family members involved in providing care will increase accordingly and supporting them is crucial in order to increase the quality of life of persons with dementia. Although there have been advances in the treatment of dementia diseases, caregiving is still a challenging situation for the family. Across Europe, family caregivers are confronted with significant challenges regarding supportive care needs for persons with dementia living at home [2, 3]. These include personal care, housekeeping, logistics and financial problems, being responsible for medications, and contact with the healthcare authorities [4]. Such a caregiving role leads to a significant responsibility and little free time for the family caregiver to take care of themselves [5]. A meta-analysis focusing on stress and coping for family caregivers to persons with dementia shows that being a family caregiver is a significant risk factor which can lead to decreased physical health and increased psychological stress [6]. It is well known that family caregiving for persons with dementia is associated with negative consequences for the caregivers such as care burden [7], stress, strain [8], sadness, depression, and anxiety [9], feelings of loneliness and social isolation [10–12] and reduced quality of life [13]. Social isolation and lack of formal support have a negative impact on family caregivers' personal and social life [10]. Evidence indicates a strong association between being a family caregiver of a person with dementia and symptoms of depression [14, 15]. Lower perceived health of family caregivers is associated with an increased risk for the institutionalisation of persons with dementia [16]. Even in terms of physical health, it has been reported that it is easier to be a caregiver to persons with other chronic diseases compared to persons with dementia [17].

The various types of support given to family caregivers to counteract their stress, burden, lack of subjective well-being and social isolation include organising support groups, education about dementia, social work services and face-to-face conversations [18, 19]. Different technical solutions can provide an alternative way and a platform for family caregivers to obtain support from care professionals or to exchange experiences with other family caregivers who are in similar situations [20]. The use of app-based technologies in smartphones is rapidly growing in all age groups. Mobile health (mHealth) applications have increased throughout the world and have been developed for a wide variety of health interventions, predominantly as mobile applications (app) for self-care such as for asthma [21] and diabetes [22]. mHealth is defined

**Funding:** he author(s) received no specific funding for this work.

**Competing interests:** The authors have declared that no competing interests exist.

as "medical and public health practice supported by mobile devices, such as mobile phones, patient monitoring devices, personal digital assistants and other wireless devices" [23]. Older people are more likely to use technology that they are already familiar with [24]. For older people, their motivation and perceived relevance of the interventions are important determinants of the use of mHealth tools [25–27]. Other important factors for using mHealth interventions include their usability and accessibility [27]. Compared to face-to-face meetings with family caregivers, the technical solutions available in the home provide a tool to connect with care professionals without leaving home. mHealth tools are always available and accessible, and they are especially useful for people living in rural areas because of the distance they may be forced to travel to obtain healthcare services [28]. mHealth-based interventions have emerged as a potential solution to support family caregivers and address some of the challenges related to caregiving. Studies have shown that technology-based interventions can reduce depression [6, 29], anxiety, and stress [29, 30] which may lead to improved health and wellbeing of the family caregivers [6, 31–33]. Recent research from Sweden and Denmark, conducted by our research group, on expectations of family caregivers of persons with dementia about a mobile app through which they could be supported by care professionals reported that it would help caregivers to be recognized as individuals needing support [34, 35]. The family caregivers also mentioned that they should receive the support as and when they needed it, at their own pace. The aim of the present study was to describe the experiences of family caregivers of persons with dementia who received professional support through a mobile app and usage of the app.

## Method

### Study design

This qualitative descriptive study describes the experience of receiving professional support through the interactive mobile app STAV [" STöd för AnhörigVårdare" (Support for family caregivers)] by family caregivers of persons with dementia living at home.

### Intervention

Support was offered to family caregivers of persons with dementia (henceforth referred to as family caregivers) by two nurses of the research team through a mobile app over an eight-week period. The mobile application is unique as it was developed by involving the stakeholders, family caregivers and health care professionals and because of it is interactive nature. When developing the app, family caregivers were asked about the contents they would like to have in such a mobile application. The healthcare professionals were asked which types of care needs could be addressed through a mobile application or if it could facilitate communication with caregivers. Both the family caregivers and professionals suggested several functions which would facilitate support in addressing challenging caregiving situations [34]. The mobile app STAV has the following features: 1) chat, 2) mindfulness exercises, 3) weblinks to relevant sites, 4) own contact list, and 5) personal diary [36]. These features were meant to encourage dialogue with care professionals (via chat), to take care of oneself as a caregiver (by using mindfulness exercises), to get an overview of relevant information (collection of web links), and by taking own notes in the diary to record challenging situations to discuss with the care professionals at a later stage. Family caregivers could also create their own contact list of services relevant to their caregiving tasks. The mobile app can be downloaded on a smartphone or tablet that uses the operating systems Android or iOS. A log-in with a username and a pin code is required to use the app.

## Participants

Fifteen family caregivers of persons with dementia were recruited to test STAV. Family caregivers who accompanied a family member with dementia to the outpatient cognitive clinic at a major hospital in Stockholm were approached to participate in the study and were selected consecutively. Inclusion criteria of the family caregivers of persons with dementia were that the persons with dementia would live at their own home, the family caregivers were 18 years or older, they were able to read and write Swedish, and they had their own mobile device (smartphone or tablet) with their own internet connection. Family caregivers who were physically or cognitively challenged to communicate were excluded from the study. A total of 12 family caregivers agreed to use the app and to be interviewed. Of the participating family caregivers, seven were women and five men (one was a child to the person with dementia and 11 were partners).

## Data collection

The interviews were conducted 4–16 weeks after the use of the mobile app. Although the interviews were meant to take place immediately after the eight-week long support by the nurses in the research team, in late January 2020, data collection was interrupted due to the restrictions related to Covid19 in Sweden. This also resulted in 10 interviews having to be conducted over the telephone instead of face to face. The initial two interviews were conducted face to face at the participants' homes.

A semi-structured interview guide was developed with the specific purpose of exploring the experiences of using the mobile app and its different features. The interviews were between 12–51 minutes long and conducted by one of the co-authors of the study (APK). The respondents were asked about the relevance of each of the features and ease of use of the mobile app. The questions in the semi-structured interview guide with family caregivers were open yet focused on the aim of the study. For example, family caregivers were asked what they thought about the app, their experiences of using the app, and what they thought about the specific features of the app. All the interviews were recorded with permission from the participants.

## Data analysis

All the recorded interviews were transcribed verbatim. The interview text was analysed using Braun & Clarke's [37] six-step thematic analysis approach. The first step was to become familiar with the data. The transcriptions were read and re-read by three of the authors (ÅD, HK, ZNK) several times, taking notes and marking initial ideas. In the second and third steps, coded extracts from the entire data set were independently generated by the three and grouped into potential themes (see example of the analysis process in Table 1). In the fourth step, the three co-authors (ÅD, HK, ZNK) refined the themes together and produced a thematic

**Table 1. Example of the data analysis.**

| Data | Extraction of code | Theme |
|---|---|---|
| Yes, exactly, I also felt it felt safe and. . .yes it was helpful to know that there was someone, somewhere to get in touch with as well | Support | Filling a gap |
| At the same time, I thought mindfulness was really good, you sometimes stress yourself out completely unnecessarily and try to unwind—it benefits everyone. | Calming | Foundation for inner calm |
| I wrote down all of my worries, it was a lot of worries and there was so much then everything and I thought it was awful, I wrote and wrote | Diary | Way of offloading |

overview. In the fifth step, the same co-authors continued to name and define the themes together. The last step was to produce the report based on the analysis. All co-authors read and commented on the results.

## Ethical considerations

Written informed consent was obtained from the participants of the feasibility study to test the intervention of providing professional support using STAV. Verbal and recorded informed consent was taken prior to conducting the qualitative interviews of the current study. Participants were also reminded that they could withdraw from the study at any point, that the data provided by them would be kept confidential and assured anonymity. The study was approved by the Swedish Ethical Review Authority (Ref.: 2018/1160-31/5).

## Results

As a result of the analysis, six themes emerged. The themes were named *Filling a gap*, *Right time and right place*, *Foundation for inner calm*, *Better introduction and overcoming technical barriers*, *Relevant information in one place*, and *Way of offloading*.

## Filling a gap

This theme highlights the need for a mobile app for family caregivers. The general experience with the chat and the interaction with the nurse was positive.

> *"Good contact point with experts"* (Interview 3)

The family caregivers also said that they would recommend the app to others if it was available. The chat feature allowed them to get in touch with a professional if needed.

> *"It was helpful to know that there was someone, somewhere"* (Interview 9)

Most family caregivers experienced that they received a good response from the nurse when posing a question.

> *"I got ideas and tips about how to handle the future and it was good."* (Interview 8)

They said that the chat was the best feature as it provided the possibility to have direct contact with a health professional at their own convenience. This gave a sense of security to the family caregivers.

> *"It felt like having a friend during troubled times."* (Interview 6)

Several caregivers requested to be part of a group where peers could communicate with each other. They missed the possibility to join a group chat where they could share experiences and get support from others in similar situations if they were sad or felt lost.

> *"It would have been valuable to exchange some experiences with each other."* (Interview 10)

Some family caregivers valued a group chat but wanted to be anonymous in the chat. However, not all family caregivers were interested in participating in a group chat with other caregivers but would appreciate the opportunity to join one if they felt differently with time. It was

described as difficult to discuss personal issues without compromising the privacy and dignity of the family member with dementia. The family provided all the support that was needed and there was no need for support from peers in a group chat.

*"I will never discuss my wife's situation with anyone else outside the family."* (Interview 2)

Some family caregivers indicated that they did not use the app so much but would use the mobile app in the future if it were available and if the situation of their family member with dementia deteriorated. Although they did not necessarily use the app it felt reassuring to have it, to know that someone was on the other end.

## Right time and right place

This theme discusses the need for a moderator to get the chat to function as support for the family caregivers. It also refers to the possibility for the caregivers to use the mobile app when they have the time and need it most. It entailed no stress, there was no appointment to be met by the family caregivers, it was flexible to use whenever they needed to. Some family caregivers described that they used the app when they were alone. Having the mobile app during the Covid-19 pandemic was a good alternative to face-to-face meetings due to the restrictions about physical distancing.

*"Felt good during Corona to have the chat, it is a good idea, this must continue, to provide chat"* (Interview 8)

The app was not needed during the entire time the person with dementia was cared for by the family caregiver. Rather, the need for support varied depending on the progression of the disease. The family caregivers described that when they first started testing the app, they did not use it so much, but it became of greater value later when the disease had progressed and the need for information, advice, and support increased.

*"Yes, I missed it afterwards... So, I did not use it that much just during the test period, but I noticed it* (its absence) *because my husband has recently been diagnosed so it is worse with each passing day, it would probably have been needed now."* (Interview 3)

The app was a good contact point with experts.

*"Great support, I felt a support, from the health care, from you. When I was home alone with the problems, I had to ask and could get answers pretty quickly."* (Interview 3)

The nurses played an important role as a moderator to stimulate the family caregivers to use the chat feature on the mobile app. The caregivers felt that they needed someone else to take the initiative to engage in a dialogue.

*"Need someone else (*other than themselves*) to take initiative to start a dialogue"* (Interview 1)

Receiving short messages was appreciated by the family caregivers and to get answers directly so they did not have to wait for the next doctor's appointment.

*"Information and prompt answers in small quantity was good"* (Interview 4)

The possibility to chat individually with the nurse through the mobile app was experienced as positive. It was good to have the opportunity to ask questions. The family caregivers described it as valuable to get quick answers to their questions. However, some caregivers experienced that they did not get answers from the nurse in time or it was too sporadic, so they stopped using the chat.

## Foundation for inner calm

This theme relates to the individual experience of using the mindfulness feature in the mobile app. It worked as a foundation for inner calm for the family caregivers when they were in a mood to relax. They needed a starting point of inner calm to take advantage of this feature. If they had time and space for themselves, the mindfulness feature helped to relax. The family caregivers experienced the feature as good and calming. It reminded them to take time for themselves, to sit down, listen and unwind. This is when they had their own time and could focus solely on themselves.

> *"Mindfulness was really good, you sometimes stress yourself out completely unnecessarily and try to unwind—it benefits everyone."* (Interview 3)

Some of the caregivers were a little sceptical to the mindfulness feature, the reason being the lack of time for them to wind down. Some mentioned that it was not suitable in a home with a person with dementia, that they could not relax that much. Limited time and a lack of energy were described as obstacles in using the mindfulness feature. Family caregivers also mentioned that they could not use it in the right way, it did not give them the calm and stillness it suggested. However, they had an interest in exploring the mindfulness feature further if time permitted.

> "The mindfulness features were well thought out, I am very happy with it." (Interview 6)

The mindfulness session was a good length so they could keep their focus on it.

## Better introduction and overcoming technical barriers

This theme encapsulates the need family caregivers have for an introduction to the mobile app STAV. Most caregivers had a positive response to the mobile app, but they felt they needed a better introduction about its purpose, the relevance of each feature, and how it could benefit the users.

> *"I probably did not understand its purpose clearly, what I should do, because I had a few problems in the beginning."* (Interview 4)

They also requested better information about who they were chatting with. Being introduced to the nurse was important to feel secure when discussing personal emotions and problems.

> *". . ...and the question is who I am sharing it* (the problems) *with then*?" (Interview 9)

The family caregivers felt that the mobile app needed to be simple, user-friendly, and easy to navigate. Using the mobile app was not a problem for most of the family caregivers given their technical know-how, but there were certain technical difficulties with the mobile app itself which created problems for the users, necessitating a restart of the app several times.

*"It was difficult to get into the app but was easy to use."* (Interview 8)

## Relevant information in one place

The theme addresses the value of having all available information sources in one place. Family caregivers said it was good to have relevant information collected in the mobile app which was a way of keeping oneself updated and receiving useful information that related to one's specific situation.

"*This* (the links) *has given me what I need in principle.*" (Interview 11)

Through the weblinks, the family caregivers got information about the different associations and organisations available for dementia care. The information also gave ideas and support on how to handle the future and the progression of the diseases. Some of the caregivers stated that the links were the best feature of the mobile app.

"*The app provides security, I can look for answers to questions I have and find the help I need, so it helps me in this way.*" (Interview 6)

## Way of offloading

The theme focuses on the diary in the mobile app as a way of documenting issues related to caregiving for the family member with dementia and have it as a basis for discussion with care professionals. It was good to have the notes so as not to forget points of discussion. It was also used as a timeline for the progress of the disease as it is difficult to notice small changes in a person with dementia when together all the time. The diary also served as a platform to write down own thoughts where they wrote down their worries.

"*I wrote down my worries, there was so much then, so I wrote and wrote.*" (Interview 8)

It was a way to lift the burden from their shoulders.

## Discussion

The results of this study provide insight into experiences, needs, and the barriers faced by family caregivers of persons with dementia in receiving professional support through the mobile app STAV and its usage. The main findings show that there is a need for a mobile app for family caregivers to receive support and information easily and without having to leave home to meet health care professionals. This is in line with studies that describe technical solutions as useful for those who are unable to leave their family member with dementia unattended for a while [38, 39]. Digital-based interventions offer family caregivers the opportunity to access the information at their convenience in terms of time and from the privacy of their own homes [40]. In the present study, the family caregivers had the opportunity to use it when they had the time and at their own pace. The interactivity of the app was appreciated by the family caregivers and most said they would recommend the app to others. Most apps are designed for information seeking, medical advice, advice for self-management, or designed for advanced care planning of patients with chronic conditions [41]. To the best of our knowledge, there are no scientific reports describing how such interactive apps were developed and evaluated. Rathnayake et al. [42] found in their study that caregivers had a positive attitude towards the mHealth app as being useful and addressing information and caregivers' educational needs. It is essential for family caregivers to get suggestions as to what they as caregivers can do to

provide better care for a person with dementia [43]. The results from this study show that the mobile app STAV fills a gap for the family caregivers and its flexibility is appreciated by the caregivers. They can use it whenever it is convenient for them. The feeling of a sense of security was an experience many caregivers shared, to know that there was a person on the other end ready to help if needed. The chat feature was described as the best part of the app. The family caregivers had mixed reactions about the possibility of a group chat. Some family caregivers lacked the opportunity to have a group chat and wanted to share experiences with peers in a similar situation. Studies describe peer support as a key component of support for family caregivers of persons with dementia. Specifically, support received from other participants in an e-health or m-health programme in the form of discussing emotions, having supportive messages, sharing information, and motivation are key factors to bring about positive effect on caregiver stress [44].

The family caregivers in the current study had varying needs of the app and it was mainly based on the progression of the disease of the family member and the time available for oneself. The interaction with the nurse through STAV was valued. Caregivers expressed that the app had the potential to provide support when the nurses were active and took contact with the family caregivers. This is in line with a study where the health care professional had a proactive approach and contacted the family caregivers which significantly decreased hardship and grief compared to the control group [45]. Overall, the contact with the nurse was a positive experience for the family caregivers. Some of the family caregivers mentioned challenges in the interaction with the nurse, such as the nurse was not always proactive, the time between asking a question and receiving a response was sometimes too long. Some described a lack of continuity not knowing who the contact nurse was, which created feelings of confusion and insecurity. Other research also mentions that facilitating factors in communication with health care professionals are motivation, support, and feedback from the staff [25].

One caregiver mentioned he had no interest in using the group chat feature in the app because it was not the right way to obtain support. His own family was a better source of support. Family and friends are an important factor for family caregivers of persons with dementia [46]. Family support and reflections of their situation are particularly important when life situation changes and their roles in the family changes from being partner or children to caregiver [47].

An expressed need by the family caregivers in the present study was to be better informed about the objective and relevance of the app, more information about each of the features, and when to use the different features. The family caregivers mentioned that the app was easy to use although some technical problems were faced during the test period. The contents in the app were considered relevant by most family caregivers. The web links in the app were perceived as an easy and accessible source of information. Using the app for support provides the caregivers with a sense of being connected to the health team which can lead to more effective support for caregivers [48].

## Strengths and limitations

To strengthen the credibility of this study, three of the co-authors discussed with each other at each step of the analysis process and reached a consensus regarding the themes. The three co-authors also independently reflected upon the results in relation to the interview texts to ensure that nothing was overlooked. A limitation in this study was that the data collection could not be conducted directly after the intervention due to the COVID-19 pandemic. The lapse of time between the end of the intervention to data collection spanned over a couple of weeks to four and a half months which could have led to some recall bias. All participating

family caregivers were supposed to be interviewed face-to-face but due to the restrictions on physical distancing during the COVID-19 pandemic, some of the interviews were conducted via telephone. This affected the length of the interviews, the telephone interviews being significantly shorter than the two face-to-face interviews. Although interviews on distance are often shorter than face-to-face interviews, both have shown to contain the same range of topics [49]. Telephone interviews may also have affected how much the family caregivers expressed about their experiences over the telephone. It is reported that telephone interviews can give the participants a feeling of confidence to share their inner thoughts [50]. A limitation of the current study was that one of the nurses providing support through the app was the same person responsible for the follow-up interviews with the family caregivers. This could potentially have inhibited the family caregivers in sharing their experiences frankly.

## Conclusion

Interactive apps for family caregivers of persons with dementia have the potential of being an important tool through which to provide professional support. This is particularly of relevance given "ageing in place" policy and digitalisation of health and social care services in Sweden. This app has a core that is the same for all but can be modified to provide support to other FCs providing care to persons with other chronic conditions. The chat feature enabled direct communication at the time and place when the caregiver needed it. This especially filled a gap during the COVID 19 pandemic when face to face meetings were not possible.

Our research indicated that care professionals have to take an active role in the chat forum to simulate interaction with the family caregivers. The mindfulness feature was used to facilitate selfcare of the family caregivers. The app enabled caregivers to access information and support when it was convenient for them. However, the users need careful introduction to the use of the app and help to overcome technical barriers. Further research is needed to assess the feasibility of the app as an intervention tool for family caregivers to persons with dementia and also to assess it is effectiveness on relieving impact of caregiving.

## Supporting information

**S1 File.**
(DOCX)

## Author Contributions

**Conceptualization:** Hanne Konradsen, Zarina Nahar Kabir.

**Data curation:** Åsa Dorell, Ana Paula Kallström, Zarina Nahar Kabir.

**Formal analysis:** Åsa Dorell, Hanne Konradsen, Ana Paula Kallström, Zarina Nahar Kabir.

**Methodology:** Åsa Dorell, Hanne Konradsen, Zarina Nahar Kabir.

**Project administration:** Zarina Nahar Kabir.

**Supervision:** Hanne Konradsen, Zarina Nahar Kabir.

**Writing – original draft:** Åsa Dorell, Hanne Konradsen, Zarina Nahar Kabir.

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
