## [Decision Letter · Decision Letter 0]

21 Sep 2021

PONE-D-21-18728“A friend during troubled times”: Experiences of family caregivers looking after people with dementia when receiving professional support via a mobile appPLOS ONE

Dear Dr. Dorell,

Thank you for submitting your manuscript to PLOS ONE. After careful consideration, we feel that it has merit but does not fully meet PLOS ONE’s publication criteria as it currently stands. Therefore, we invite you to submit a revised version of the manuscript that addresses the points raised during the review process.

It is requested to (see the enclosed reviewers' comments):

-include more and longer quotations presenting the words of the interviewee;

-provide more information or references about the STAV app;

- carefully discuss the "gaps of knowledge" behind this work in relation to the relevant literature;

-include a table that displays the sub-themes that emerged, in addition to just listing the six themes;

- revise the declaration of intent in the abstract;

- better specify the narrative in the introduction;

- revise the conclusion adding future directions and perhaps the specificities of the Swedish case;

- carefully revise the language;

-add a few lines explaining the differences between the terms "dementia" and "Alzheimer";

-harmonize the presentation of numbers in the manuscript.

We look forward to receiving your revised manuscript.

Kind regards,

Filomena Papa

Academic Editor

PLOS ONE

Journal Requirements:

3. Please include additional information regarding the interview guide used in the study and ensure that you have provided sufficient details that others could replicate the analyses. Please include a copy, in both the original language and English, as Supporting Information.

Reviewers' comments:

Reviewer's Responses to Questions

**Comments to the Author**

1. Is the manuscript technically sound, and do the data support the conclusions?

Reviewer #1: Partly

Reviewer #2: Yes

2. Has the statistical analysis been performed appropriately and rigorously? 

Reviewer #1: N/A

Reviewer #2: N/A

3. Have the authors made all data underlying the findings in their manuscript fully available?

Reviewer #1: Yes

Reviewer #2: Yes

4. Is the manuscript presented in an intelligible fashion and written in standard English?

Reviewer #1: Yes

Reviewer #2: Yes

5. Review Comments to the Author

Reviewer #1: I do NOT have any potential or perceived competing interests that may influence my review. For further certainty, I state that I have NO actual or potential pecuniary, professional, reputational or other interests that might influence or might be perceived to influence my review.

In the interest of transparency, the Comments to the Author are identical to those provided to the Editor.

Thank you for the opportunity to review this manuscript, which presents the results of a qualitative study of a mobile application designed to support caregivers of people with dementia. The subject (dementia) is perennially important, it addresses a substantial clinical need (caregiver strain), and it involves a novel intervention (mHealth app). It is clearly written and easy to follow. It appears to conform to reporting guidelines.

I entertain some concern that the size of the study (12 interviewees) may not be large enough to justify publication in a large journal of general interest, but I will leave this matter to the Decision Editor.

I recommend that the manuscript be resubmitted with MAJOR REVISIONS.

In general, the ratio of analysis to evidence is too high. I would like to see more and longer quotations to let the interviewees "speak for themselves" a bit more. I think richer quotations would be helpful in illuminating and in some cases supporting your findings, and in allowing the reader to critically assess your conclusions.

As an example, on page 7, the first 2 paragraph state many different observations about multiple aspects of the theme but then this is only supported by basically 2 sentences of quoted text.

Particularly as your study about the basis for your app is unpublished, I would also like to know a lot more about STAV, how it was developed (and any research that it was informed by), what you think is unique about it, and so on. It's a bit hard to get a sense of how the app works and how exactly it addresses perceived needs.

You also state that "not much attention has focused on the physical and mental health of the family members who have to look after their loved ones who have dementia and who take care of them at home." I think this statement needs to be qualified a bit. There are in fact at least three fairly recent reviews of interventions focused on dementia caregivers; Gitlin and colleagues state that there have been around 200 such interventions to date.

Gitlin et al. Gerontologist 2015;55:210–26

Walter et al. Gerontologist 2019;20:1–11

Hinton et al. BMJ Glob Health. 2019 Nov 12;4(6):e001830

Lastly, one specific point -- I would suggest you include a table that displays the subthemes that emerged, in addition to just listing the six Themes. This would be useful in signposting the discussion for the reader.

Reviewer #2: The paper introduces a qualitative study with caregivers of persons with dementia living at home using a mobile app.

I appreciated the article, but also found the necessity for the following revisions.

1. More information or references should be provided about the STAV app.

ABSTRACT

2. The abstract mentions "to evaluate the app's usability". I found this one a rather emphatic declaration, whereas the paper only discusses a few considerations about usablity. I suggest softening the declaration of intent.

INTRODUCTION

3. Some references report studies in specific populations (e.g. ref.2 in Sweden, ref.3 in Europe,...), yet the first paragraph of the introduction is very generic and does not account for geographic differences. I suggest a better level of specification in the narrative.

METHOD

4. At the beginning of the "Intervention" section, the sentence "an overview of available

information channels (collection of web links)" needs a verb.

5. At the beginning of the "Data collection" section, the sentence "it had to be put on hold" should be rephrased as "they had to be put on hold".

6. At the beginning of the "Ethical considerations" section, the sentence "Written informed consent were obtained" should be rephrased as "Written informed consent was obtained".

7. Similar mistakes appear here and there in the text: I suggest a careful langiage revision.

8. I am curious about the chats between nurses and caregivers via the app. Could you provide examples of topics and themes, and the level of detail they were dealt with?

CONCLUSION

9. I felt the "Conclusion" section was rather meager and generic. I suggest to improve it by also adding future directions, and perhaps the specificities of the Swedish case (if any).

10. I am not an expert of neurobiology, therefore I found it difficult to navigate through the terms "dementia" and "Alzheimer", both used in the paper. I guess it could be useful for readers like me to add a few lines explaining the differences.

11. Throughout the paper numers are sometimes written as digits and sometimes as words: please harmonise.

6. PLOS authors have the option to publish the peer review history of their article (what does this mean?). If published, this will include your full peer review and any attached files.

Reviewer #1: **Yes: **Thom Ringer

Reviewer #2: **Yes: **Bartolomeo Sapio

---

## [Author Response · Author response to Decision Letter 0]

26 Jan 2022

Include more and longer quotations presenting the words of the interviewee: See response to reviewer 1’s comment below.

Provide more information or references about the STAV app: See response to reviewer 1 & 2s’ comments below.

Carefully discuss the "gaps of knowledge" behind this work in relation to the relevant literature: The background has been revised to highlight the necessity of providing support to family caregivers of persons with dementia through mHealth solutions.

Include a table that displays the sub-themes that emerged, in addition to just listing the six themes:See response to reviewer 1’s comment below.

Revise the declaration of intent in the abstract: The abstract has been revised.

Better specify the narrative in the introduction: This is done following the reviewers’ advice.

Revise the conclusion adding future directions and perhaps the specificities of the Swedish case;

- carefully revise the language;

-add a few lines explaining the differences between the terms "dementia" and "Alzheimer";

-harmonize the presentation of numbers in the manuscript.:

 See response to reviewer 1 & 2s’ comments below.

Please include additional information regarding the interview guide used in the study and ensure that you have provided sufficient details that others could replicate the analyses. Please include a copy, in both the original language and English, as Supporting Information: The interview guide both in the Swedish and English is added.

Reviewer #1 

1. I would like to see more and longer quotations to let the interviewees "speak for themselves" a bit more. I think richer quotations would be helpful in illuminating and in some cases supporting your findings, and in allowing the reader to critically assess your conclusions. We have added more quotations. However, as mentioned in the paper, interviews from distance are often shorter than face-to-face interviews but contains the same range of topics. 

2. Particularly as your study about the basis for your app is unpublished, I would also like to know a lot more about STAV, how it was developed (and any research that it was informed by), what you think is unique about it, and so on. We have expanded the description of development oft the intervention in the method section. We have also referred to the publication that described how the proposed intervention was to be developed (Kabir et al, 2020).

3. You also state that "not much attention has focused on the physical and mental health of the family members who have to look after their loved ones who have dementia and who take care of them at home." I think this statement needs to be qualified a bit. The sentence is rewritten. We thank the reviewer for the references on interventions for family caregivers. One of these references has been incorporated in the introduction describing different types of support for family caregivers.

4. I would suggest you include a table that displays the subthemes that emerged, in addition to just listing the six Themes. This would be useful in signposting the discussion for the reader.

 A table on data analysis process has been included. As described in the section on data analysis, we did not include subthemes as part of the analytic process. We refer to Braun & Clark’s (2006) article in describing the steps of analysis undertaken in the study.

Reviewer #2: 

1. More information or references should be provided about the STAV app.

 We have expanded the description of development of the intervention in the method section. We have also referred to the publication that described how the proposed intervention was to be developed (Kabir et al, 2020).

2. The abstract mentions "to evaluate the app's usability". I found this one a rather emphatic declaration, whereas the paper only discusses a few considerations about usablity. I suggest softening the declaration of intent. It is changed in the manuscript and the abstract by using the word ‘usage’ instead of ‘usability’ which is a more technical word as the reviewer indicates.

3. Some references report studies in specific populations (e.g. ref.2 in Sweden, ref.3 in Europe,...), yet the first paragraph of the introduction is very generic and does not account for geographic differences. I suggest a better level of specification in the narrative.

 The introduction now begins with the global picture followed by the European situation on challenges faced by family caregivers in caring for persons with dementia. Thereafter we provide examples of specific challenges for caregivers across countries. 

4. At the beginning of the "Intervention" section, the sentence "an overview of available

information channels (collection of web links)" needs a verb. It is corrected.

5. At the beginning of the "Data collection" section, the sentence "it had to be put on hold" should be rephrased as "they had to be put on hold".

 It is reformulated.

6. At the beginning of the "Ethical considerations" section, the sentence "Written informed consent were obtained" should be rephrased as "Written informed consent was obtained". It is corrected.

7. Similar mistakes appear here and there in the text: I suggest a careful language revision.

 The manuscript has been thoroughly checked for linguistic errors.

8. I am curious about the chats between nurses and caregivers via the app. Could you provide examples of topics and themes, and the level of detail they were dealt with?

 An example of the topic of the chat between the family caregiver and the researcher included how to address behavioral challenges of the person of dementia. However, the chat data was not saved for analysis as we did not have ethical approval for it. 

9. I felt the "Conclusion" section was rather meager and generic. I suggest to improve it by also adding future directions, and perhaps the specificities of the Swedish case (if any). The conclusion has been revised to capture the implications of the specific findings of our study.

10. I am not an expert of neurobiology, therefore I found it difficult to navigate through the terms "dementia" and "Alzheimer", both used in the paper. I guess it could be useful for readers like me to add a few lines explaining the differences. We have chosen to use the generic term dementia so as not to confuse the readers.

11. Throughout the paper numbers are sometimes written as digits and sometimes as words: please harmonise. We have followed the following rule: a number is written in words when a sentence begins with a number; in case of single digital number, it is written in words; and in case of larger numbers than single digit they are written as numeric. The manuscript has been checked for mistakes.

---

## [Decision Letter · Decision Letter 1]

2 May 2022

PONE-D-21-18728R1“A friend during troubled times”: Experiences of family caregivers to persons with dementia when receiving professional support via a mobile appPLOS ONE

Dear Dr. Dorell,

Thank you for submitting your manuscript to PLOS ONE. After careful consideration, we feel that it has merit but does not fully meet PLOS ONE’s publication criteria as it currently stands. Therefore, we invite you to submit a revised version of the manuscript that addresses the points raised during the review process.

It is requested to revise the manuscript according to comments of Reviewer 2.

We look forward to receiving your revised manuscript.

Kind regards,

Filomena Papa

Academic Editor

PLOS ONE

Journal Requirements:

Additional Editor Comments (if provided):

The use of pictures (e.g. screen shots, drawings) could be useful to describe the STAV app.

Reviewers' comments:

Reviewer's Responses to Questions

**Comments to the Author**

1. If the authors have adequately addressed your comments raised in a previous round of review and you feel that this manuscript is now acceptable for publication, you may indicate that here to bypass the “Comments to the Author” section, enter your conflict of interest statement in the “Confidential to Editor” section, and submit your "Accept" recommendation.

Reviewer #2: (No Response)

Reviewer #3: All comments have been addressed

2. Is the manuscript technically sound, and do the data support the conclusions?

Reviewer #2: Yes

Reviewer #3: Yes

3. Has the statistical analysis been performed appropriately and rigorously? 

Reviewer #2: N/A

Reviewer #3: N/A

4. Have the authors made all data underlying the findings in their manuscript fully available?

Reviewer #2: Yes

Reviewer #3: Yes

5. Is the manuscript presented in an intelligible fashion and written in standard English?

Reviewer #2: Yes

Reviewer #3: Yes

6. Review Comments to the Author

Reviewer #2: Although the authors improved the paper by addressing the reviewers' comments, I must confess I am not satisfied with the ways they chose to address some of those comments.

The reviewers asked for more information about the STAV app, how it was developed (and

any research that it was informed by), what you think is unique about it, and so on. The authors added a generic sentence bearing little or no extra information.

The reviewers asked to improve conclusions by adding future directions and the specificities of the Swedish case. The authors slightly modified one sentence without really tackling the issue.

The reviewers asked to include a table that displays the subthemes that emerged. The authors added a table with a few examples of the data analysis, which I do not find helpful at all.

I suggest the authors consider the reviewers' comments in a more serious way, if they want to get their paper published in the journal. These are not major revisions and can be done with a little effort.

Reviewer #3: (No Response)

7. PLOS authors have the option to publish the peer review history of their article (what does this mean?). If published, this will include your full peer review and any attached files.

Reviewer #2: **Yes: **Bartolomeo Sapio

Reviewer #3: **Yes: **Merle Varik

---

## [Author Response · Author response to Decision Letter 1]

20 Jun 2022

Thanke you for your comments, we have revised the manuscript after the reviewer #2 comments.

---

## [Decision Letter · Decision Letter 2]

12 Jul 2022

“A friend during troubled times”: Experiences of family caregivers to persons with dementia when receiving professional support via a mobile app

PONE-D-21-18728R2

Dear Dr. Dorell,

We’re pleased to inform you that your manuscript has been judged scientifically suitable for publication and will be formally accepted for publication once it meets all outstanding technical requirements.

Kind regards,

Filomena Papa

Academic Editor

PLOS ONE

Additional Editor Comments (optional):

All comments have been addressed. Thank you.

Reviewers' comments:

Reviewer's Responses to Questions

**Comments to the Author**

1. If the authors have adequately addressed your comments raised in a previous round of review and you feel that this manuscript is now acceptable for publication, you may indicate that here to bypass the “Comments to the Author” section, enter your conflict of interest statement in the “Confidential to Editor” section, and submit your "Accept" recommendation.

Reviewer #2: All comments have been addressed

Reviewer #3: All comments have been addressed

2. Is the manuscript technically sound, and do the data support the conclusions?

Reviewer #2: Yes

Reviewer #3: Yes

3. Has the statistical analysis been performed appropriately and rigorously? 

Reviewer #2: N/A

Reviewer #3: N/A

4. Have the authors made all data underlying the findings in their manuscript fully available?

Reviewer #2: Yes

Reviewer #3: Yes

5. Is the manuscript presented in an intelligible fashion and written in standard English?

Reviewer #2: Yes

Reviewer #3: Yes

6. Review Comments to the Author

Reviewer #2: Thanks for addressing my comments: tha paper is now ready for publication.

Reviewer #3: This qualitative descriptive study's aim is achieved, and the manuscript was well and appropriately revised.

7. PLOS authors have the option to publish the peer review history of their article (what does this mean?). If published, this will include your full peer review and any attached files.

Reviewer #2: **Yes: **Bartolomeo Sapio

Reviewer #3: No

---

## [Editor Report · Acceptance letter]

20 Jul 2022

PONE-D-21-18728R2 

*“A friend during troubled times”:* Experiences of family caregivers to persons with dementia when receiving professional support via a mobile app 

Dear Dr. Dorell:

I'm pleased to inform you that your manuscript has been deemed suitable for publication in PLOS ONE. Congratulations! Your manuscript is now with our production department. 

Kind regards, 

on behalf of

Dr. Filomena Papa 

Academic Editor

PLOS ONE